# Tissue-Based Markers as a Tool to Assess Response to Neoadjuvant Radiotherapy in Rectal Cancer—Systematic Review

**DOI:** 10.3390/ijms23116040

**Published:** 2022-05-27

**Authors:** Edgaras Smolskas, Goda Mikulskytė, Ernestas Sileika, Kestutis Suziedelis, Audrius Dulskas

**Affiliations:** 1Department of Abdominal Surgery, Vilnius City Clinical Hospital, 57 Antakalnio Str., LT-10207 Vilnius, Lithuania; edgaras.smolskas@gmail.com; 2Faculty of Medicine, Vilnius University, LT-08412 Vilnius, Lithuania; goda.mikulskyte@gmail.com; 3Department of Radiotherapy, National Cancer Institute, 1 Santariskiu Str., LT-08406 Vilnius, Lithuania; ernestas_sileika@yahoo.com; 4Department of Abdominal and General Surgery and Oncology, National Cancer Institute, LT-08660 Vilnius, Lithuania; kestutis.suziedelis@nvi.lt; 5Life Sciences Center, Institute of Biosciences, Vilnius University, LT-08412 Vilnius, Lithuania

**Keywords:** rectal cancer, neoadjuvant chemoradiotherapy, radiosensitivity markers, micro-RNA, tumor immune microenvironment

## Abstract

According to current guidelines, the current treatment for locally advanced rectal cancer is neoadjuvant therapy, followed by a total mesorectal excision. However, radiosensitivity tends to differ among patients due to tumor heterogeneity, making it difficult to predict the possible outcomes of the neoadjuvant therapy. This review aims to investigate different types of tissue-based biomarkers and their capability of predicting tumor response to neoadjuvant therapy in patients with locally advanced rectal cancer. We identified 169 abstracts in NCBI PubMed, selected 48 reports considered to meet inclusion criteria and performed this systematic review. Multiple classes of molecular biomarkers, such as proteins, DNA, micro-RNA or tumor immune microenvironment, were studied as potential predictors for rectal cancer response; nonetheless, no literature to date has provided enough sufficient evidence for any of them to be introduced into clinical practice.

## 1. Introduction

Rectal cancer (RC) makes up about one-third of all colorectal cancer (CRC) cases worldwide, with 0.7 million cases reported in 2020. By 2040, the number is likely to increase to up to 1.16 million cases per year [1]. According to the current guidelines of the National Comprehensive Cancer Network (NCCN) for Rectal Cancer, the current treatment for locally advanced RC is neoadjuvant therapy (NT), followed by a total mesorectal excision (TME) [2,3]. The American Society of Colon and Rectal Surgeons (ASCRS) recommends using NT for patients diagnosed with clinical stage II or III RC [4]. As several studies have shown, approximately 10–14% of RC patients achieve pathologic complete response (pCR) following preoperative therapy, which has also been associated with better disease-free survival (DFS) and tumor recurrence [5,6]. As an alternative to TME, a watch-and-wait strategy is a possible approach for those patients who achieve pCR after NT [7]. However, radiosensitivity tends to differ among patients due to tumor heterogeneity, making it difficult to predict the possible outcomes of the NT [8,9]. Moreover, there is no reliable method to date that can estimate how the tumor will respond on an individual patient basis.

Several previous publications have attempted to determine the possible correlation between various pre-NT factors and pCR. During a digital rectal examination, tumors defined as fixed rather than mobile, as well as tumors located low (<4 cm) or high (>8 cm) in the rectum, are less likely to achieve pCR [10,11]. Likewise, tumors proctoscopically seen as stenotic or involving more than 90% of the circumference tend to respond poorly to NT [9]. In addition, lower clinical stages of T and N, as well as lower tumor grades, higher doses of radiation and longer periods between NT and surgery were independent factors associated with a higher likelihood of achieving pCR [12]. The potential of using imaging modalities, such as magnetic resonance imaging and positron emission tomography–computed tomography, is also explored in numerous studies as a possible predictor of pCR [13,14,15,16]. However, the sensitivity and specificity of these clinicopathological and imaging features are seemingly low. Although there is no unanimity of independent predictive factors in the achievement of pCR in RC patients so far, molecular biomarkers, either tissue-based or blood-based, appear to have the potential to predict the possible outcomes of the NT with adequate accuracy [17,18].

This review aims to investigate different types of tissue-based biomarkers and their capability of predicting tumor response to NT in patients with locally advanced rectal cancer (LARC).

## 2. Results

### 2.1. Biopolymers of Cancer Cells

#### 2.1.1. Proteins

Numerous protein biomarkers have been studied in past years to better understand their role in the possible RC response to radiotherapy. These include DNA-binding protein SATB1 (Special AT-rich sequence binding protein, SATB1), X-ray repair cross-complementing protein 2 (XRCC2), Human phosphatidylethanolamine-binding protein 4 (hPEBP4), Cytoplasmic phosphatidylinositol transfer protein 1 (PITPNC1), Forkhead box proteins K1 and K2 (FOXK1, FOXK2), Apoptosis regulator Bcl-2 (B-cell lymphoma 2, Bcl-2), Cyclooxygenase-2 (Cox-2), Vascular endothelial growth factor (VEGF), Apoptotic protease activating factor-1 (APAF-1), Fibroblast growth factor 8 (FGF8), Fibroblast growth factor receptor 4 (FGFR4), Survivin, Focal adhesion kinase (FAK), Golgi phosphoprotein 3 (GOLPH3), PCNA-associated factor (Proliferating cell nuclear antigen-associated factor of 15 kDa, PAF15), Beclin-1, Class II Nuclear factor-kappaB (subunit p65) (N-κB/p65), Polo-like kinase 1 (PLK1), Ataxia telangiectasia mutated (ATM), Double-strand break repair protein MRE11 (Human homolog of Meiotic recombination 11, MRE11), Pantetheinase (Vascular non-inflammatory molecule-1, VNN1), Serine/threonine-protein kinases VRK1 and VRK2 (Human vaccinia-related kinases 1 and 2, VRK1 and VRK2), human homologous recombination proteins RAD51 and RAD52 (RAD51, RAD52), p53 binding protein 1 (53BP1) and Tumor protein p53 (p53).

SATB1 is a nuclear matrix-associated protein that is located in the human chromosome 3p23 [19]. In several studies by Menge et al., the authors showed that levels of SATB1 were significantly higher in tumor tissue compared to normal rectal tissue (*p* = 0.043). Moreover, knocking down SATB1 in cell lines and then exposing them to 2Gy radiation statistically significantly increased the sensitivity to radiotherapy when compared to the control cell lines (*p* < 0.001) [20,21]. In relation to radiation-associated factors, SATB1 appeared to have a positive correlation with Ki-67 and Survivin, and a negative correlation with ataxia telangiectasia mutated (ATM) and pRb2/p130 (*p* < 0.05) [21]. Furthermore, in a study by Lopes-Ramos et al., SATB1 expression seemed to increase when the expression of miR-21-5 was inhibited, decreasing the radiosensitivity in the cell lines [22].

The nuclear protein XRCC2, located on the 7q36.1 chromosome, and other somatic RAD51 paralogs (RAD51B, RAD51C, RAD51D and XRCC3) are responsible for Deoxyribonucleic Acid (DNA) repair and chromosome stability. When cells become deficient in XRCC2, it leads to chromosome instability and increased cell sensitivity to ionizing radiation [23,24]. The Qin et al. study of 67 patients with locally advanced rectal cancer demonstrated that cancer cells with knockdown of XRCC2 (sh-XRCC2) became more sensitive to ionizing radiation. To further investigate this increase in radiosensitivity, changes in biochemical markers of apoptosis before and after ionizing radiation were identified, with increased levels of poly adenosine diphosphate ribose polymerase (PARP), cleaved caspase-3 and cleaved caspase-9 in the sh-XRCC2 and a significant decrease in Bcl-2 when compared with the control cells [25].

Several studies have shown that the inhibition of apoptosis induced by tumor necrosis factor-α (TNF-α) and the upregulation of hPEBP4 plays a vital role in tumor progression in various types of cancer [26,27,28,29]. Qiu et al. investigated hPEBP4 as an independent biomarker for predicting rectal cancer response to radiotherapy treatment (*p* = 0.001). Tumors with a higher expression of hPEBP4 appeared to be more resistant to radiotherapy, and the patients had worse progression-free survival when compared to those with a low expression of hPEBP4 [30]. When hPEB4 was inhibited with a chemical inhibitor of hPEBP4 IOI-42, it resulted in increased radiosensitivity in rectal cancer cells in vivo [30].

PITPNC1 has also been linked to the metastatic progression of several types of cancer, including CRC [31,32]. Tan et al. discovered a significantly higher expression of PITPNC1 in radioresistant rectal cancer tissues than in radiosensitive tissues. Moreover, the overexpression of PITPNC1 seemed to lower the production of reactive oxygen species (ROS), thus increasing the radioresistance in rectal cancer cells [33]. However, it is notable that the study only included biopsies from 16 patients.

FOXK1 and FOXK2, members of the FOXK protein family, are known to regulate many processes of the cell, including starvation-induced atrophy, proliferation and differentiation of the cell, cellular metabolism, autophagy and DNA repair. Therefore, dysregulations of FOXKs may contribute to decreased radiosensitivity [34]. In the study by Zhang et al., biopsies from 169 patients with LARC have shown that the expression of FOXK1 and FOXK2 was comparably higher in patients with non-pCR compared to the pCR group. Furthermore, the 3-year OS rate indicated that patients with overexpressed FOXKs in their pre-neoadjuvant chemoradiotherapy (nCRT) tumor tissues had worse OS (56.9% in the FOXK1 group and 64.2% in the FOXK2 group, *p* < 0.01) than patients with low levels of FOXK1 and FOXK2 (93.1% and 87.0%, respectively, *p* < 0.01) in their pre-nCRT biopsies [35].

Fibroblast growth factors are polypeptides that are responsible for regulating proliferation and differentiation of the cell, its survival and other biological functions [36]. In a recent study, Harpain et al. discovered low levels of FGF8 expression in four out of five complete responders [37]. A total of 89% of the patients with low levels of FGF8 in their tumor tissue were responders, whereas in the group with high levels of FGF8, only 44% responded well to the nCRT (*p* = 0.003). A total of 87% of patients with low levels of Survivin expression achieved good response to nCRT; at the same time, only 50% of the patients with high levels of Survivin in their tumor tissue responded well to the treatment (*p* = 0.02). Interestingly, out of the five complete responders, only two had low levels of Survivin [36]. Survivin belongs to the gene family responsible for the inhibition of apoptosis, and its upregulation can be found in most types of cancer. High levels of Survivin also take part in the proliferation and angiogenesis of cells, making it an important part in the formation of cancer [38]. In a retrospective study of 116 patients with LARC conducted by Yu et al., the expression levels of Survivin in the tumor tissue were analyzed. Positive immunostaining for nuclear Survivin was observed in 30 of the patients (25.9%), and 33 patients (28.4%) showed positive immunostaining for cytoplasmic Survivin. Positive nuclear (*p* = 0.001) or cytoplasmic (*p* = 0.003) Survivin correlated with a worse 5-year DFS, suggesting a possible correlation between the overexpression of Survivin in the tumor tissue and radioresistance in RC patients [39].

FGFR4 is a transmembrane tyrosine kinase receptor. It has been discovered that patients showing a strong response or complete response to nCRT have a significantly lower expression of FGFR4 in their tumor tissue, whereas patients with high FGFR4 expression correlated with poor radiosensitivity (*p* = 0.04) [40].

The cytoplasmic protein FAK is responsible for the regulation of cell signaling and survival of the cancer cell [41]. In a study done on 73 patients diagnosed with locally advanced rectal adenocarcinoma, the expression levels of FAK were significantly lower in non-responders when compared to responders (*p* = 0.007) [42].

PAF15 is a small PCNA-associated protein mainly localized in the mitochondria and nucleus of the cell. [NO_PRINTED_FORM]Although the molecular mechanism is not yet clear, PAF15 appears to be involved in cell survival and DNA repair, possibly through the p21 and p33 pathways [43]. PAF15 expression has shown a possible correlation with tumor radiosensitivity in RC patients. Yan et al. conducted a study in which PAF15 expression in 105 biopsies of paired primary RC and normal rectal tissues was evaluated. PAF15 seemed to inhibit DNA damage that was caused by gamma irradiation, as well as promote cell proliferation in RC cells after radiation (*p* < 0.05), indicating that cellular radiosensitivity in RC could be increased by the inhibition of PAF15 [44].

Peripheral membrane phosphoprotein GOLPH3 has been identified as an oncogenic protein in several solid tumors, including colon cancer [45]. It has been previously demonstrated that GOLPH3 was overexpressed in CRC tissue when compared to normal colorectal mucosae (*p* < 0.01) and appeared to be a potential predictor of 5-FU chemosensitivity in CRC patients [46,47]. Zhu et al. found that a low expression of GOLPH3 in RC tissue correlated with a better response to nCRT. A total of 62% of patients (44 out of 71) with low expression levels of GOLPH3 showed tumor down-staging after nCRT; in contrast, only 43% (33 out of 77) with GOLPH3 overexpression in tumor tissue showed down-staging after nCRT (*p* = 0.020) [48].

Beclin-1, an essential Bcl-2-interacting autophagy protein, is a negative regulator of tumor formation and mammalian cell growth [49]. Zaanan et al. discovered that patients with LARC who had high Beclin-1 expression in their tumor tissue were less likely to achieve pCR after nCRT when compared to patients with low levels of Beclin-1 (*p* = 0.02). These patients also appeared to be less likely to show down-staging after nCRT (*p* = 0.02) [50].

The nuclear factor-kappaB (NF- κB) family is comprised of five transcription factors: p50/p105 (NF-κB1), p52/p100 (NF-κB2), p65 (RelA, RelB and c-Rel) [51]. In a study conducted by Voboril et al., the expression levels of Class II transcription factors (NF-κB/p65) in RC tumor tissue before and after nCRT were evaluated. Although the expression levels of NF-κB/p65 seemed to be higher in rectal adenocarcinoma compared to healthy tissue, NF-κB/p65 expression levels did not correlate with tumor radiosensitivity [52].

The expression of mitotic regulator PLK1 has been previously described as an important factor in the survival of cancer cells; furthermore, the overexpression of PLK1 has been discovered in various types of cancers, including colon cancer [53]. A lower PLK1 expression appears to correlate with worse pathologic response in RC patients receiving nCRT. In a study by Cebrian et al., pre-treatment tumor tissues from 75 rectal cancer patients were analyzed. Of patients who showed a high expression of PLK1, 54.2% achieved pCR or partial response, whereas only 37% with a low PLK1 expression achieved pCR or partial response after nCRT (*p* = 0.049). Low expression levels of PLK1 were also linked with reduced DFS (*p* = 0.06) [54].

Ho et al. evaluated the correlation between the expression of MRE11/ATM two-protein panel and tumor radiosensitivity in patients with RC. It was noted that patients with MRE11/ATM overexpression had worse DFS (*p* = 0.028) and OS (*p* = 0.024) [55]. Furthermore, in a more recent study, Ho et al. also assessed the MRE11/RAD50/NBS1 (MRN) three-protein panel as a potential molecular biomarker to predict radiosensitivity in RC patients. Patients with higher expression levels of the MRN complex proteins had significantly worse DFS (*p* = 0.024) and OS (*p* = 0.028) [56].

VNN1 is an ectoenzyme that is highly expressed in tissues with a high turnover of CoenzymeA (CoA), including liver, kidney and intestine, with intestinal VNN1 being predominantly expressed by enterocytes [57]. Chai et al. observed an association between a poor response to nCRT and VNN1 overexpression in 172 patients with primary rectal adenocarcinoma. High VNN1 expression levels were significantly associated with worse disease-specific survival (*p* = 0.0001), as well as with worse local recurrence-free survival (*p* = 0.0001) and poor response to nCRT (*p* = 0.001) [58].

In a study conducted by Peng et al., the predictive value of APAF-1 and COX-2 expression in tumor tissue taken from patients with LARC was evaluated. Patients with a low expression of COX-2 and a high expression of APAF-1 had the highest pCR rate of 56% when compared to low COX-2 expression/low APAF-1 expression (17.4%), high COX-2 expression/low APAF-1 expression (15.4%) and high COX-2 expression/high APAF-1 expression (14.3%) (*p* = 0.05) [59].

RAD51 recombinase is a DNA repair protein that plays an important role in homologous recombination, which is crucial for the maintenance of a normal cell cycle. Thus, any dysregulation in the process of DNA repair may lead to carcinogenesis and cancer progression [60]. In a study on RAD51 overexpression in colorectal adenocarcinomas, Tennstedt et al. examined a subgroup of patients with rectal cancer and discovered that the overexpression of RAD51 increased tumor resistance to radiotherapy [61]. Recombination protein RAD52 also plays an important role in DNA repair by maintaining stabilization of the fork and stopping excessive fork reversal [62]. In a study of 179 patients, 40 underwent preoperative treatment and were examined for a possible connection between the preoperative expression of RAD52 and patients’ response to the treatment. A lower expression of RAD52 was associated with worse DFS and overall survival and increased resistance to radiotherapy [63].

The cellular protein 53BP1 binds to the central domain of p53 [64]. As Huang et al. discovered in a recent study, patients with a higher expression of 53BP1 in their pretreatment biopsy tissue were more likely to achieve a better response to nCRT than patients with lower levels of 53BP1 expression [65].

The p53 protein, as a potential biomarker, has been investigated both due to the functions of this protein in the cell after DNA damage as well as the frequent mutations in the gene (TP53) of this protein in different human cancers. The mutational status of TP53 adds additional complexity; therefore, we reviewed the attempts to assess the potential of p53 as a biomarker in the following chapter of this review.

#### 2.1.2. Genetic Markers—Mutations and Expression of Protein Coding Genes

Different gene mutations are being considered as potential markers to predict the outcome of neoadjuvant therapy for rectal cancer patients. The most commonly mutated genes in colorectal cancer are the Adenomatous polyposis coli (APC), Tumor protein p53 (TP53), and Kirsten rat sarcoma viral oncogene homolog (KRAS) genes, but there are many other gene mutations that, alone or in combination, could be used to predict the rectal cancer response to radiotherapy or chemoradiotherapy.

KRAS is a proto-oncogene located on chromosome 12p12.1. KRAS becomes oncogenic due to point mutations at codons 12 and 13 and less frequent mutations, including those in codon 61 [66]. Mutations of the KRAS gene are found in about 50% of colorectal cancer cases. There are conflicting data with regards to the predictive significance of KRAS mutations in RC. In 2011, Garcia-Aguilar et al. reported that rectal cancers with a KRAS mutation are less likely to develop a pCR to neoadjuvant CRT compared to tumors with wildtype KRAS [67]. After two years, they published another study showing that mutations in different KRAS codons may have different effects on rectal cancer resistance to CRT [68]. In 2015, Martellucci et al. also found that patients with a KRAS codon 13 mutation were more likely to be resistant to nCRT than other patients and were less likely to achieve pCR, whereas mutations in codons 12, 6 and 61 did not significantly affect pCR [69]. Peng et al. researched the correlation between different mutated oncogenes and clinical outcomes in locally advanced RC and revealed that tumors with KRAS mutations responded poorly to preoperative chemoradiotherapy (*p* = 0.044) [70]. Gaedcke et al. isolated DNA from pre-therapeutic biopsies of 94 patients with rectal cancer treated with preoperative chemoradiotherapy in order to establish the mutation status of KRAS exons 1–3. The treatment response levels were compared between patients without a KRAS mutation and those with a mutation in either codons 12, 13, 61 or 146. In contrast to previous studies, none of these comparisons showed a significant difference between the groups [71].

Another quite extensively studied gene is the p53 protein gene, also known as Tumor protein p53 gene, TP53. This gene codes for a protein that regulates the cell cycle and, hence, functions as a tumor suppressor. The loss or mutation of TP53 is linked to an increased risk of cancer [72]. However, the results of various studies investigating TP53 and mutant or wt p53 levels as a potential biomarker to predict rectal cancer response to nCRT differ, as the results from various studies tend to contradict [73]. Lopez et al. found that, although 50% of the patients had mutations in TP53, no correlation between changes of p53 in cancer tissue and different responses to radiotherapy was found [74]. Another study on the Moroccan population found a high expression of p53 in tumor tissue in 93% (39) of patients with an incomplete response and in only 7% of complete responders, demonstrating that p53 may be of prognostic value in the therapeutic response of LARC [75]. Similarly, Hur et al. found that 46.9% of patients with low-expressed p53 in their tumor tissue achieved pCR, whereas only 24.5% of patients with overexpressed p53 were able to achieve pCR [76]. Hence, it should be noted that TP53 mutational status was not evaluated in either of these studies, and this may have significant importance for evaluating the potential of p53 as a biomarker. Another two studies found an association between mutant TP53 and tumor response to radiation [77,78].

Interestingly, the previously mentioned study by Garcia-Aguilar et al. revealed that mutant TP53 alone is not associated with tumor response to CRT, but a TP53 mutation together with a KRAS gene mutation was identified as a possible predictor of a non-pCR to CRT [67].

Few studies have focused on microarray analysis using rectal cancer tissues to identify gene expression profiles associated with the response to radiotherapy or chemoradiotherapy. Although the predictive potential of microarray data has looked promising, only an overlap of the Filamin A (FLNA) and Matrixmetallopeptidase 14 (MMP14) genes was observed between the gene lists announced in these studies [79,80,81,82]. This could be explained by differences in the tumor contents, chemotherapy regimens, microarray platforms or analytical tools.

In recent years, Douglas et al. published a study in which a genetic variation in rectal cancer patients with complete response to chemoradiation versus poor response were compared. They investigated potential mutations in the genetic profiles of rectal cancer patients before and after nCRT. Mutations in Lysine demethylase 6A (KDM6A), Non-receptor tyrosine kinase ABL1 (ABL1), variations of the Death domain associated protein–Zinc finger-BTB domain containing protein 22 complex (DAXX-ZBTB22) and KRAS genes were only found in poor responder samples (not mutated in complete responders). Ten genes, including AT-Rich Interaction Domain 1A (ARID1A), Mismatch Repair System Component PMS1 Homolog 2 (PMS2), Janus kinase 1 (JAK1), CREB binding protein (CREBBP), Mammalian target of rapamycin protein kinase (MTOR), RB transcriptional corepressor 1 (RB1), Protein kinase cAMP-dependent type I regulatory subunit alpha (PRKAR1A), F-box/WD repeat-containing protein 7 (FBXW7), variations of ATM-Chromosome 11 open reading frame 65 (ATM C11orf65) and Lysine methyltransferase 2D (KMT2D), were only mutated in the patients who were complete responders to neoadjuvant chemoradiation. However, because of the relatively small sample size of the study (*n* = 16 patients), it is not possible to make reliable conclusions [83].

Zauber et al. assessed a series of rectal cancers for the molecular changes of loss of heterozygosity in the APC and Netrin receptor DCC (DCC) genes, K-ras mutations, and microsatellite instability. None of the molecular changes were useful as indicators of regression [84].

In a study conducted by Gantt et al., biopsies from a total of 33 patients with rectal cancer of the middle-third or lower-third were taken [85]. The difference in the expression of 19228 genes was analyzed in responders (AJCC score 0–2) and non-responders (AJCC score 3). The top 10 genes that were upregulated in non-responders were Apolipoproteins A-I, A-II, B and CIII (APOA1, APOA2, APOB, APOC3), Alpha-2-HS-glycoprotein (AHSG), Dopamine beta-hydroxylase (DBH), LIM homeobox transcription factor 1 alpha (LMX1A), Sterol O-acyltransferase 2 (SOAT2), Transferrin (TF) and Solute Carrier Family 7 (SLC7A9). The downregulated genes in non-responders included LOC729399, Serine incorporator 5 (SERINC5), Sodium channel non-voltage-gated 1 beta subunit (SCNN1B), Zinc finger CCH-type containing 6 (ZC3H6), Solute carrier family 4 sodium bicarbonate cotransporter member 4 (SLC4A4), DTW domain containing 2 (DTWD2), Membrane-spanning 4-domains subfamily A member 1 (MS4A12), Brain expressed X-linked 5 (BEX5), Multimerin 1 (MMRN1) and Chloride Channel Accessory 4 (CLCA4) [85].

In a study on various hub genes associated with RC response to CRT by Sun et al., the authors discovered that pCR was more likely to be achieved if the expression of hub genes Decorin (DCN) and collagen type XV alpha 1 chain (COL15A1) was higher. However, integrin beta 1 (ITGB1), Notch receptor 3 (NOTCH3) and Secreted protein acidic and cysteine rich (SPARC) were more likely to be overexpressed in patients with non-pCR. Furthermore, DCN, ITGB1, NOTCH3 and SPARC genes were discovered to be strongly associated not only with tumor response to nCRT but also with disease-free survival; they were further used to form a four-gene expression-based risk score [86].

The overexpression of two lipid biosynthesis-associated genes, 17β-hydroxysteroid dehydrogenase type 2 (17HSD2) and mitochondrial enzyme 3-hydroxy-3-methylglutaryl-CoA synthase (HMGCS2), was associated with poor response to concurrent chemoradiotherapy treatment in the biopsies of 46 rectal cancer patients. Moreover, the upregulation of these genes was linked to worse tumor regression grade, disease-free survival and metastasis-free survival [87]. The enzyme 17HSD2 is expressed in the epithelium of the colon, as well as in the epithelial cells of the stomach, small intestine and urinary bladder. It is mainly responsible for the conversion of estradiol to estrone and testosterone to androstenedione [88].

#### 2.1.3. Micro-RNA

Micro-RNAs (miRNAs) are endogenous non-coding RNAs that negatively regulate genes’ expression by pairing to their targeted messenger RNA (mRNA) [89]. By targeting tumor suppressor genes (TSG), miRNAs can behave as oncogenic miRNAs or act as tumor-suppressing miRNAs if the target is an oncogene [90].

Lopes-Ramos et al. analyzed the expression profile of miRNA in biopsies taken before nCRT in 27 patients with rectal adenocarcinomas (cT2-4 N-0 M-0) and compared it in patients with complete and incomplete responses to nCRT. The overexpression of miR-21-5, miR-1246 and miR-1290 was seen in complete responders, while the overexpression of miR-205-5p was noted in the group of incomplete responders. In addition, the following analyses displayed 78.5% sensitivity and 86% specificity in miR-21-5p expression and the ability to predict complete response to nCRT [22].

Carames et al., in the study of 92 LARC cases, also noted that, although there was no association between the overexpression of miR-21 and tumor grade before pre-CRT, miR-21 overexpression was significantly associated with pathological response to CRT (*p* = 0.013) [91]. Another study by Carames et al. investigated 82 patients with LARC and found that high levels of miR-31 correlated with poor outcomes in pathological response to nCRT and with worse OS (*p* = 0.008) [92].

D’Angelo et al. found that miR-194 was significantly overexpressed in responders compared to non-responders (*p* = 0.016) [93]. Hotchi et al. investigated the role of miRNAs in response to CRT in three different parameters (histopathological examination, RECIST and downstaging). Based on histopathological examination of the biopsy, miR-223 and miR-142-3p appeared to be more expressed in the responder group than in non-responders (*p* = 0.026). Based on RECIST, miR-223 was expressed at higher levels in responders than in non-responders (*p* = 0.034); meanwhile, eight genes seemed to be under-expressed in the responder group, including miR-20b (*p* = 0.048), miR-92a (*p* = 0.024), let-7a (*p* = 0.048), miR-20a (*p* = 0.041), miR-17 (*p* = 0.012), miR-106a (*p* = 0.024), mir-17 (*p* = 0.024) and miR-20-a (*p* = 0.041). Based on downstaging, miR-223 (*p* = 0.006), miR-630 (*p* = 0.042) and miR-126 (*p* = 0.049) were seen to be expressed more in responders’ cancer tissue than in non-responders. As miR-223 appeared to have a higher expression in all three parameters, it was further evaluated and chosen as a potential new tissue biomarker for predicting tumor response to CRT [94]. 

In the study by Svoboda et al., the expression levels of eight miRNAs significantly differed between non-responder and responder groups. MiR-215, miR-190b and miR-29b-2 were overexpressed in non-responders, and let-7e, miR-196b, miR-450a, miR-450b-5p and miR-99a in responders. Using these eight miRNAs, nine of ten responders and nine of ten non-responders (*p* < 0.05) were correctly classified [95].

A recent study by Baek et al. analyzed 65 tissue samples taken before CRT from patients diagnosed with LARC. A good response to CRT significantly correlated with the upregulation of three miRNAs: miR-199a/b-3p (*p* < 0.0001), miR-199b-5p (*p* < 0.0001) and miR-199a-5p (*p* = 0.0011). A higher expression of miR199a/b-3p (*p* < 0.001), miR-199b-5p (*p* < 0.001) and miR-199a-5p (*p* = 0.001) in the tumor tissue also appeared to correlate with statistically better OS [96].

### 2.2. Immunological Markers—Blood-Based and Tissue-Based

Several publications have provided evidence of tumor microenvironment involvement in modulating tumor response to chemoradiotherapy.

In addition to the already reviewed biomarker potential of 53BP1, Huang et al. discovered that tumors with highly expressed 53BP1 also had more significant T cell infiltration in comparison to tumors with a lower expression of 53BP1. Moreover, the immunoscore of CD3/CD8 was also significantly higher in tumors with a high 53BP1 expression [66]. Yasuda et al. evaluated the density of T lymphocytes in 48 RC biopsies and tumor response to CRT. A correlation between higher numbers of CD3(+) T cells, CD8(+) T cells and increased tumor radiosensitivity was noted. Based on a barium enema study, a decrease in tumor size was strongly correlated with the density of CD4(+) T cells (*p* = 0.0013), as well as the density of CD8(+) T cells (*p* = 0.0020) [97].

Tumor-associated macrophages (TAMs) play an important role in tumor progression by inhibiting immune responses and promoting angiogenesis [98]. It appears that TAMs affect the tumor microenvironment by suppressing cytotoxic T lymphocyte responses [99]. In a study of 191 patients with LARC, macrophage-associated biomarkers CD163, CD68, macrophage colony-stimulating factor (MCSF) and C-C chemokine ligand 2 (CCL2) in pre-nCRT and post-surgery tumor tissue were found to be higher in patients who were associated with a poorer response to nCRT and lower in patients with a complete pathological response [100]. A study of 85 patients revealed that CD26 overexpression in rectal cancer cells was correlated with a poor pathological response to CRT. Moreover, it was noted that patients with high levels of CD26 were more likely to have serosal and vascular invasion [101].

#### 2.2.1. Blood-Based Immunological Markers

A study by Caputo investigating the neutrophil-to-lymphocyte (N/L) ratio before and after neoadjuvant chemoradiotherapy treatment in patients with rectal cancer showed that the N/L ratio prior to treatment was a predictor of poor tumor treatment response [102]. An elevated N/L ratio after the treatment was, however, associated with a worse outcome. Another study looked at the N/L ratio of tumors after neoadjuvant treatment but prior to surgery and found that poor responders had a significantly higher value of N/L after neoadjuvant therapy compared to good responders [103].

Some studies indicate that cytokine levels (TNF-α, IL-6, CD40L, CCL-5, TGF-β1) in patient serum could be useful, but analyses of multiple cytokines most likely need to be combined for a more general clinical applicability, and any such indicator must be further validated [104,105,106].

#### 2.2.2. Tissue-Based Immunological Markers

The type, quantity and location of tumor-infiltrating lymphocytes (TILs) have been thoroughly investigated in pre- and post-treatment tissue biopsies of rectal cancer with the aim to predict the response to treatment. Recently, lymph-node ratio (LNR) has emerged as a prognostic tool and is known as a predictive marker for survival in rectal cancer.

#### 2.2.3. Immunological Biomarkers within Tumors

In recent years, the B7 family has received increased attention. The B7 family contains checkpoint molecules that regulate immune responses by providing positive signals for T cell growth, differentiation and cytokine production. At least ten B7 family members have been identified: CD80 (B7-1), CD86 (B7-2), PD-L1 (B7-H1), PD-L2 (B7-DC or CD273), ICOSL (B7-H2), CD276 (B7-H3), B7S1 (B7-H4, B7x or Vtcn1), VISTA (B7-H5, GI24 or PD-1H), B7-H6 and B7-H7 (HHLA2) [107]. In 2020, Wang and colleagues retrieved and analyzed the TCGA database and reviewed three B7 family molecules, including B7-H3, VISTA and HHLA2, as the most expressed in patients with colorectal cancer. They introduced these checkpoint molecules as a potential immunotherapeutic target for patients with colorectal cancer [108]. Some other studies found that B7-H3, B7-H4 and B7-H7 expression in colorectal cancer was significantly upregulated as compared with normal tissues. The high expression of these molecules was correlated with a poor outcome in patients with colorectal cancer [109,110,111,112]. In 2020, Yasui et al. published a study whose aim was to investigate the effect of chemoradiotherapy on the immunological status of rectal cancer patients who were treated with preoperative chemoradiotherapy. The results demonstrated that the expression of immune checkpoint genes, such as B7-H3 and B7-H5, was upregulated after chemoradiotherapy [113]. Therefore, it can be supposed that B7 family checkpoint molecules could be used as predictors of response to chemoradiotherapy in colorectal cancer patients.

However, further larger studies are needed.

### 2.3. Other

Other reviewed factors included microbiota and cancer markers (CEA).

The gut microbiota have been identified as a potentially important factor in how cancer responds to therapies, and there have been attempts to modulate it to yield more favorable outcomes [114,115].

## 3. Discussion

There are specific molecular differences among patients with different responses to NT (Table 1). However, it is worth mentioning that most of the studies reviewed in this article include a single type of tissue-based molecular biomarker. Due to molecular heterogeneity in RC patients, it is unlikely that a single molecular marker with sufficient sensitivity and specificity in predicting NT response would be identified. Therefore, it is particularly important not only to examine these types of biomarkers in comparison to each other but also to integrate other clinicopathological and imaging modalities into the same sample sets. This approach could potentially benefit the further development of a more reliable biomarker model.

Some limitations in the currently available data on the predictive value of various molecular biomarkers are worth mentioning. The small sample sizes included in a study may impact the reliability of given results; therefore, larger cohort studies are necessary to achieve more trustworthy results. In addition, factors such as variability in treatment schedules and different NT regimens could influence findings in the currently available studies. Future studies should also include a standardized evaluation system of tumor response, allowing a better comparison and interpretation of the results.

Many molecular biomarkers are studied as potential predictors for RC response; nonetheless, no literature to date has provided sufficient evidence for any of them to be introduced into clinical practice. It is possible that incorporating different combinations of molecular biomarkers into the same sample sets may offer additional specificity and sensitivity, which is lacking when the marker is studied independently of others. The integration of molecular biomarkers into clinical practice could be beneficial for predicting tumor response and for further personalization in the care of rectal cancer patients.

## 4. Materials and Methods

We performed this systematic review according to the Preferred Reporting Items for Systematic Reviews and Meta-Analyses (PRISMA) statement [116].

### Literature Search and Inclusion Criteria

Two authors independently searched the electronic databases of the Cochrane Library, Embase, Web of Science, CENTRAL and PubMed until 30^th^, September 2021. The search strings used in Medline and Embase were: “rectal”, “cancer”, “predictive”, “response”, “prediction”, “predictor”, “biomarkers”, “tissue”, “neoadjuvant”, “radiotherapy”, “chemoradiotherapy” and “neoadjuvant treatment”. Single words and different search combinations were used.

This review focuses on studies published in the English language between January 1995 and September 2021.

The inclusion criteria were as follows: (1) original studies; (2) studies that analyzed predictors for the response to treatment of rectal cancer with neoadjuvant radiotherapy; and (3) patients who underwent treatment only for rectal cancer.

Exclusion criteria were as follows: literature reviews, papers with limited information, articles on molecular biomarkers of colorectal cancer and articles or abstracts written in non-English.

Response to NT was evaluated based on pCR, partial response, tumor regression grading (TRG), overall survival (OS), Response Evaluation Criteria in Solid Tumor (RECIST) and American Joint Committee on Cancer (AJCC) score.

Any disagreement was solved by consensus or by a third reviewer. Data from included studies were extracted into a datasheet and pretested to prove their suitability. In addition, references and abstracts were searched. We identified 96 abstracts in NCBI PubMed and selected 41 reports considered to meet the inclusion criteria (Figure 1).

Proteins in this review are named according to the recommendations available in the UniProt database (https://www.uniprot.org, last accessed 7 April 2022).

## 5. Conclusions

We have described potential biomarkers, including molecular genetic markers, immunological markers and other biomarkers, that have been analyzed from patient samples to predict the response in rectal cancer patients undergoing neoadjuvant chemoradiotherapy.

## Figures and Tables

**Figure 1 ijms-23-06040-f001:**
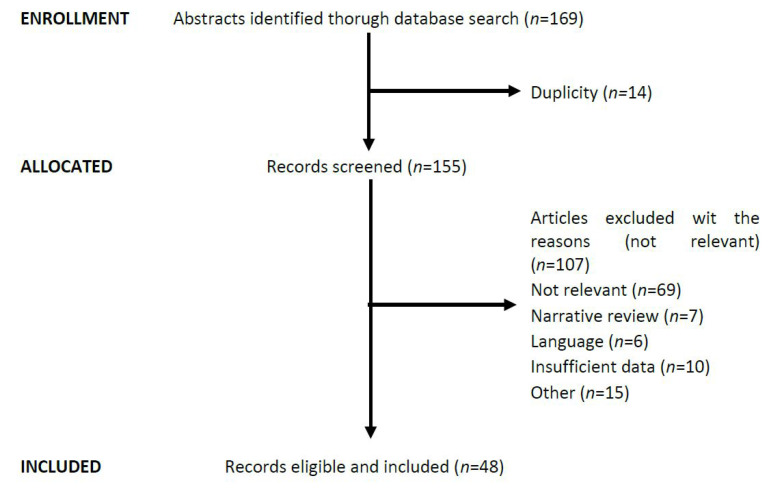
Flowchart describing the identification and inclusion of studies.

**Table 1 ijms-23-06040-t001:** Biomarkers suggested predicting neoadjuvant treatment response in rectal cancer.

Category	Type	Parameter
Biopolymers of Cancer Cells	Proteins	SATB1 XRCC2, hPEBP4, PITPNC1, FOXK1, FOXK2, Bcl-2, Cox-2, VEGF, APAF-1, FGF8, FGFR4, Survivin, FAK, GOLPH3, PAF15, N-κB/p65, PLK1, ATM, MRE11, VNN1, VRK1, VRK2, RAD51, 53BP1, PITPNC1
	Tumour suppressors and oncogenes	TP53, XIAP, TCF4, RAD51
	Transcriptome/Epigenome	Transcriptomic and epigenetic signatures, miR-21, miR-223, miR-31, miR-106a, miR-20b
Immunological markers	Tissue-based immunological markers	TIL
	Blood-based immunological markers	cytokines
	Immunological biomarkers within tumors	CD80 (B7-1), CD86 (B7-2), PD-L1 (B7-H1), PD-L2 (B7-DC or CD273), ICOSL (B7-H2), CD276 (B7-H3), B7S1 (B7-H4, B7x or Vtcn1), VISTA (B7-H5, GI24 or PD-1H), B7-H6 and B7-H7 (HHLA2)
Other biomarkers	Blood-based cancer markers, gut microbiota	-

Abbreviations: DNA-binding protein SATB1 (Special AT-rich sequence binding protein, SATB1), X-ray repair cross-complementing protein 2 (XRCC2)XRCC2), Human phosphatidylethanolamine-binding protein 4 (hPEBP4), Cytoplasmic phosphatidylinositol transfer protein1 (PITPNC1), Forkhead box proteins K1 and K2 (FOXK1, FOXK2), Apoptosis regulator Bcl-2 (B-cell lymphoma 2, Bcl-2), Cyclooxygenase-2 (Cox-2), Vascular endothelial growth factor (VEGF), Apoptotic protease activating factor-1 (APAF-1), Fibroblast growth factor 8 (FGF8), Fibroblast growth factor receptor 4 (FGFR4), Survivin, Focal adhesion kinase (FAK), Golgi phosphoprotein 3 (GOLPH3), PCNA-associated factor ( Proliferating cell nuclear antigen associated factor of 15 kDa, PAF15), Beclin-1, Class II Nuclear factorkappaB (subunit p65) (N-κB/p65), Polo-like kinase 1 (PLK1), Ataxia telangiectasia mutated (ATM), Double-strand break repair protein MRE11 (Human homolog of Meiotic recombination 11, MRE11), Pantetheinase (Vascular non-inflammatory molecule-1, VNN1), Serine/threonine-protein kinases VRK1 and VRK2 (Human vaccinia-related kinases 1 and 2 VRK1 and VRK2), human homologous recombination proteins RAD51 and RAD52 (RAD51, RAD52), p53 binding protein 1 (53BP1), Tumour protein p53 (p53).

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
