# Peer review of "Tissue-Based Markers as a Tool to Assess Response to Neoadjuvant Radiotherapy in Rectal Cancer—Systematic Review"

_ijms, 2022, doi:10.3390/ijms23116040_

Round 1

Reviewer 1 Report

ijms-1723360 titled "Tissue-based markers as a tool to assess response to neoadjuvant radiotherapy in rectal cancer-systematic review" described biomarkers for evaluating the response of rectal cancer. 

Authors analyzed various literature from several electronic databases. This study could provide information on potential predictors for researchers. Thus, I believe that the topic is of interest to the journal and that it may attract the interest of the reader. 

Minor point: 

Please, check the format of references (especially, 103-109).

Author Response

Dear Reviewer,

Thank you for your letter and constructive comments concerning our manuscript entitled “Tissue based markers as a tool to assess response to neoadjuvant radiotherapy in rectal cancer – systematic review”. The paper was revised substantially. Following changes have been made. They are as follows:

Reviewer #1:

Please, check the format of references (especially, 103-109).

References were changed according the guidelines (the changed word will be attached with all the responses to both reviewers' comments). Thank you

Thank you for valuable comments.

The manuscript quality increased following the corrections.

Sincerely

Audrius Dulskas, MD, PhD

Reviewer 2 Report

Smolskas et al. submitted a paper titled "Tissue based markers as a tool to assess response to neoadjuvant radiotherapy in rectal cancer – systematic review ". The author summarizes the role of biomarkers to evaluate the response to neoadjuvnat RT. The author provides a well-organized review article to describe the importance of biomarkers for response to RT. But the author can improve this manuscript for better understanding.

  1. Biomarkers-guided therapies have been documented for years. The authors should organized whether there are other alternative strategies to enhance the response of neoadjuvant RT.
  2. The immunological biomarkers within tumors are largely lack, such as immune checkpoint proteins PD-L1.
  3. The definition of category is inappropriate. The category "protein " should be stratified based on their classifications such as DNA repair, metabolism and so on.

Author Response

Dear Reviewer,

Thank you for your letter and constructive comments concerning our manuscript entitled “Tissue based markers as a tool to assess response to neoadjuvant radiotherapy in rectal cancer – systematic review”. The paper was revised substantially. Following changes have been made. They are as follows:

Reviewer #2:

  1. Biomarkers-guided therapies have been documented for years. The authors should organized whether there are other alternative strategies to enhance the response of neoadjuvant RT. This was not a goal of our review. However, this is a very interesting and valuable topic for future articles. 
  2. The immunological biomarkers within tumors are largely lack, such as immune checkpoint proteins PD-L1. We have added additional paragraph of your suggested immunological biomarkers. Thank you. 
  3. The definition of category is inappropriate. The category "protein " should be stratified based on their classifications such as DNA repair, metabolism and so on.  We are grateful to Reviewer for the suggestion. We have classified biomarkers in this study based on the source of the biomarker (cancer cell, tumor microenvironment, microbiota). Cancer biomarkers have been further classified using the idea of the central dogma of molecular biology explaining the flow of the genetic information. Similarly our classification is based on the level of the genetic information at which biopolymer is identified as biomarker. In case the biomarker is identified at the protein level the biomarker in this study is classified as “protein” to note that analysis at the other level of the flow of the genetic information was not performed and the potential of biomarker of the specific biopolymer is confirmed at the “protein” level. Similarly gene mutations and gene expression of protein coding genes at RNA level are classified as “genetic markers”. We agree that classification based on the function of specific molecule in cancer cell is an option, however we have chosen the different classification to note the method to be used to further exploit the potential of the specific biomarker.   

    We have slightly changed biomarker classification in the revised version of the manuscript introducing the class of “1. Biopolymers of cancer cells” and renumbering the subclass “1. Proteins” as “1.1 Proteins” to better conform the classification we have used (see attachments).

Thank you for valuable comments.

The manuscript quality increased following the corrections.

Sincerely

Audrius Dulskas, MD, PhD

This manuscript is a resubmission of an earlier submission. The following is a list of the peer review reports and author responses from that submission.